# Social Frailty Prevalence among Older People in Hong Kong

**Jed Montayre \*, Kay Kuo** [ID] **and Ka Man Carman Leung** [ID]

School of Nursing, The Hong Kong Polytechnic University, Hong Kong 999077, China;
kay-hy.kuo@polyu.edu.hk (K.K.); kmcarman.leung@polyu.edu.hk (K.M.C.L.)
\* Correspondence: jed-ray.montayre@polyu.edu.hk

**Abstract:** Background: The global increase in the ageing population underscores the importance of a holistic approach to gerontological research. Social frailty, a state of vulnerability, is a growing concern that significantly affects the well-being and health outcomes of older people. With Hong Kong projected to have the world's largest ageing population by 2050, research on social frailty within this demographic is crucial. This study aimed to assess the prevalence of social frailty among older adults in Hong Kong and examine its association with demographic characteristics. Methodology: A cross-sectional survey was conducted using data from an online survey on older adults in Hong Kong, yielding 200 respondents. The survey encompassed demographic details, the Social Frailty Scale (SFS-8), and health-related factors. Results: Participants were categorized into three groups: social non-frailty (SNF, 41.5%), social pre-frailty (SPF, 34.5%), and social frailty (SF, 24%). Spearman's rank correlation analysis revealed that self-rated health status negatively correlates with social frailty (SF) (r = −0.19, $p < 0.001$) and the number of diagnosed health conditions (r = −0.29, $p < 0.001$) but positively correlates with education level (r = 0.14, $p < 0.05$). Notably, the SPF group exhibited the highest prevalence of high cholesterol, hypertension, visual impairments, and diabetes, followed by the SF and SNF groups. No significant relationship was found between gender and SF, the total number of diagnosed health conditions and SF, or individual chronic diseases and SF. Conclusion: This survey on social frailty among older people in Hong Kong found a higher prevalence of pre-frail and socially frail individuals compared to other regions. While many benefit from strong social support, socially pre-frail and socially frail individuals have reduced interactions, highlighting the importance of social connectedness. The higher incidence of social frailty, especially among the pre-frail, underscores the need to consider Hong Kong's unique socio-cultural and economic contexts. As the first of its kind in the region, this study paves the way for further research and emphasizes the need for culturally appropriate assessment tools to better understand and address social frailty.

**Keywords:** social frailty; measurement; chronic disease; older adult; Hong Kong

## 1. Introduction

A rise in the ageing population is unfolding in countries worldwide, and research on older people's health and well-being is becoming very important [1]. Solutions to address the challenges that older people face are urgently needed. Preventing or delaying frailty is essential to resolving these issues. Frailty has phenotypes such as clinical frailty and social frailty, with both impacting older adults and being associated with one another [2]. There is an abundance of scientific literature and discussions about clinical frailty because of its measurable nature, such as its physical manifestations, mostly accompanied by cognitive function changes [3]. A decline in physical functioning results from the loss of physiological reserves and affects physical performance, with outcomes such as weakness and impaired levels of mobility [4]. On the other hand, cognitive impairment within the tenets of frailty influences changes in functions such as memory, attention, and language [5]. In contrast to physical and cognitive frailty, social frailty in older adults sometimes presents late or even goes unnoticed, primarily due to a lack of obvious external manifestations or signs of its

direct physical effects. Social frailty refers to social aspects and involves social activities (e.g., social participation, volunteering, neighbourhood involvement), social resources (e.g., family ties, care from others, presence of friends), and fulfilment of social needs (e.g., sense of belonging, emotional support, and trustful relationships) [6,7]. At its core, social frailty refers to a decline or loss of social resources and functions: social support from friends, family, or organizations; participation in social activities like seeing friends, exercising, or entertainment; and social connections in the community, like meeting neighbours or volunteering [8–10]. Although both frailty types lead to adverse outcomes among older adults, each represents distinct dimensions. Recognising these differences enables approaches to support the needs of older adults in both the physical and social domains.

Compared with clinical frailty, social frailty is a newly labelled concept that has gained attention in recent years; although it can be quickly understood as an individual's experience of personal social situations and circumstances, multiple social factors at play contribute to a socially frail status [11]. Specific factors associated with social frailty include infrequent social contact, feelings of loneliness or isolation, reduced participation in social activities, diminished self-worth, and limited social and emotional support from sources like neighbours, non-profit organizations, government services, friends, or family [12]. Studies indicate that social frailty influences individuals' physical, mental, and social abilities, thereby affecting older adults' overall health and quality of life [13]. Ringer's research [14] showed that social frailty is linked to challenges within family dynamics. When family members take on the care of socially frail older adults, they frequently experience emotional strain, as well as physical and financial burdens. This, in turn, can negatively impact the health and well-being of the caregiving family members, both physically and mentally. Some studies have indicated that social frailty is correlated with higher risks of dementia and Alzheimer's among older adults [15]. Additionally, it has been associated with negative outcomes in older adults, including disability, cognitive impairment, and mortality [16]. Furthermore, older adults identified as socially frail often reduce their social engagements (e.g., maintenance of close relationships); participate less in activities like volunteering, religious observances, work, and community involvement; and lack support from family, friends, or community members. All these factors can then promote physical decline and increase the likelihood of developing physical frailty [17]. Moreover, among vulnerable older individuals, the resulting social isolation can progress to the development of depressive symptoms, which further raise the levels of social frailty [18].

Based on the United Nations Population Division, the number of individuals aged over 65 is estimated to double by 2050, reaching 1.6 billion [19]. Asian countries will lead the trend with Hong Kong, South Korea, Japan, and Taiwan. Japan has the highest population of older adults now, at about 30.2%, and Hong Kong is expected to contribute to this ageing population trend in 2050 with estimates that 40% of the population will be over 65, indicating that one in three Hong Kong residents will be an older person [20]. Finding solutions to enable Hong Kong's older adults to lead healthy and fulfilling lives for as long as possible is a priority and is hugely affected by social factors. However, there is a lack of evidence from social frailty studies conducted in Hong Kong. While investigations of social issues relevant to ageing, such as loneliness and social isolation, have been established in the Hong Kong context, a more objective and comprehensive investigation of multi-layered and complex social issues such as social frailty has never been undertaken in Hong Kong. Therefore, this study aims to understand the prevalence of social frailty among older adults and explore the relationship between demographic characteristics and social frailty in Hong Kong.

## 2. Methods

### 2.1. Design of the Research

The current study utilized a quantitative, cross-sectional design to collect data from different individuals across various groups in a particular period. This study also examined how participants' health profiles are associated with social frailty levels. As a prevalence

study, the current study aimed to collect data on a proportion of the population who have the same or specific characteristics in a given period, which, in this case, provided insights into the extent of social frailty among older people in Hong Kong. The variables examined against social frailty are the common demographic characteristics that can be considered social determinants of health according to the earlier conceptualization of Bunt's social frailty dimensions.

*2.2. Participants, Data Collection, and Procedures*

The research team employed a convenience sampling approach to participant recruitment. The survey was distributed via email to the membership database of older adults in the community of a research centre at the host university. This study used online survey forms (Google Forms) as the data collection platform from 19 December 2023 to 11 January 2024. Participants completed the online survey independently using a computer or mobile device, without any direct support from the research team. The participants were older adults aged 50 years and above, residing in Hong Kong, and able to read and understand Chinese. They were asked to fill out a questionnaire that included demographic information, a scale measuring their social frailty status, and questions about health-related factors. A total of 200 samples were collected, comprising 49 males and 151 females. The participants' ages ranged from 50 to 86 years. The questionnaire had two sections: the first part asked about demographic information and health status, and the second part included questions on social frailty (Table 1).

**Table 1.** Social Frailty Scale—8 Items (SFS-8).

| |
|---|
| **Factor 1: Social resources** |
| 1, Do you sometimes visit your friends? |
| 2, Do you turn to family or friends for advice? |
| 3, Do you have someone to confide in? |
| **Factor 2: Social activities and financial resources** |
| 4, Do you go out less frequently compared with last year? |
| 5, Do you eat with someone at least one time in a day? |
| 6, Are you limited by your financial resources to pay for needed medical service? |
| **Factor 3: Social need fulfilment** |
| 7, Do you live alone? |
| 8, Do you talk with someone every day? |

*2.3. Measures*

2.3.1. Demographic Characteristics and Variables Examined

In addition to the general characteristics of gender, age, and education status, this study asked the participants questions such as, "Have you ever been formally diagnosed by a healthcare professional with any of the following health conditions: Diabetes, Hypertension, High Cholesterol, Hearing Problems, Vision Problems, Gout, Kidney Problems, or Others?" in order to collect data on chronic diseases and draw inferences in terms of their relationship to social frailty. The study also included a self-rated health status assessment, where participants were asked to rate their general health on a scale of 1 to 10, with 1 indicating very poor health and 10 indicating perfect health.

2.3.2. Social Frailty Scale—8 Items (SFS-8)

Social frailty is an important concept that influences the health and well-being of older people. However, there is no consensus on how to accurately measure and assess social frailty. The existing tools to measure social frailty have been largely based on social vulnerability factors, where individual items range from social connections to perceived individual vulnerability. These concepts of social frailty can be arbitrary and subjective, hence the conceptual discourse towards having a more objective approach to measure social frailty continues. The SFS-8, based on Bunt's social frailty concept [6], is one instrument the team found that has undergone psychometric testing and development. It considers a

three-factor structure involving social resources, social need fulfilment, social activities, and financial resources. This 8-item scale was used to measure social frailty in older adults in various studies, including the assessment of the impact of social frailty on health outcomes. Participants are required to answer binary "yes" or "no" questions (e.g., "Do you talk with someone every day?"). The total social frailty score is calculated by summing the affirmative answers to the relevant questions. For questions 1, 2, 3, 5, and 8, a "no" response is scored as 1 point, and a "yes" response is scored as 0 points. For questions 4, 6, and 7, a "yes" response is scored as 1 point, and a "no" response is scored as 0 points. The total score ranges from 0 to 8 points, which is used to categorize participants into three subgroups: social non-frailty (SNF; 0–1 point), social pre-frailty (SPF; 2–3 points), and social frailty (SF; 4–8 points). Our team reproduced and translated the scales from English to Mandarin with permission from the author Pek [21]. The Mandarin version of the SFS-8 scale presented a Cronbach's alpha of 0.67.

### 2.4. Statistical Analysis

In this study, SPSS 26.0 was utilized to analyse data. Interitem reliability was tested using Cronbach's alpha. Descriptive statistics were employed to analyse demographic data. The Spearman correlation was adopted to assess correlations between social frailty, age, education status, the number of diagnosed health conditions, and self-rated health status.

### 2.5. Ethical Considerations

Ethical approval was obtained from the Institutional Review Board at the Hong Kong Polytechnic University before the study was conducted (Application Number: HSEARS2023 1011007, Date: 12 October 2023). Participants were given a Participant Information Sheet (PIS) with details about the study, and informed consent was obtained.

### 2.6. Acknowledgments

The Mandarin version of the SFS-8 scale was reproduced with permission from the Institute of Geriatrics and Active Ageing (IGA), Tan Tock Seng Hospital, Singapore.

## 3. Results

### 3.1. Demographic Characteristics and Variables Examined

This study collected 200 samples, including 49 (24.5%) males and 151 (75.5%) females. The ages of the participants ranged from 50 to 86 years (M = 65.04, SD = 5.75). Regarding the participants' health conditions, 62 (31%) reported never being diagnosed with any health conditions, 110 (55%) were diagnosed with 1–2 chronic diseases, and 28 (14%) were diagnosed with 3–5 chronic diseases. Furthermore, the study asked the participants to rate their general health on a scale of 1 to 10, with 1 being very poor and 10 indicating perfect health. Sixty-six (33%) participants rated their health status below 6. One hundred and twenty-one (60.5%) older individuals rated their health status as 7–8, and thirteen (6.5%) participants gave themselves a health status rating of 9–10. In terms of social frailty, the results categorized the participants into three groups: social non-frailty (SNF), social pre-frailty (SPF), and social frailty (SF), accounting for 41.5%, 34.5%, and 24%, respectively. Table 2 presents the detailed demographic data of the participants.

**Table 2.** The demographic data of the participants (*N = 200*).

| Measure | N | % |
|---|---|---|
| Gender | | |
| Male | 49 | 24.5 |
| Female | 151 | 75.5 |
| Age (in years) | | |
| 50–54 | 10 | 5 |
| 55–59 | 14 | 7 |

**Table 2.** *Cont.*

| Measure | N | % |
|---|---|---|
| 60–64 | 68 | 34 |
| 65–69 | 73 | 36.5 |
| 70–74 | 24 | 12 |
| 75–79 | 10 | 5 |
| Above 80 | 1 | 0.5 |
| Educational Status | | |
| Primary or Below | 9 | 4.5 |
| High School | 89 | 44.5 |
| College/Associate Degree | 26 | 13 |
| Bachelor's Degree | 31 | 15.5 |
| Post-Grad Degree or Above | 45 | 22.5 |
| Numbers of Diagnosed Health Conditions (0–8) | | |
| 0 | 62 | 31 |
| 1–2 | 110 | 55 |
| 3–5 | 28 | 14 |
| Self-Rated Health Status (1–10) | | |
| 3–6 | 66 | 33 |
| 7–8 | 121 | 60.5 |
| 9–10 | 13 | 6.5 |
| Society Frailty Level (0–8) | | |
| SNF (0–1) | 83 | 41.5 |
| SPF (2–3) | 69 | 34.5 |
| SF (4–8) | 48 | 24 |

### 3.2. Correlations among Variables

The relationship between social frailty (SF) and demographics was analysed using Spearman's rank correlation analysis. Self-rated health status showed a negative association with SF ($r = -0.19$, $p < 0.001$) and the number of diagnosed health conditions ($r = -0.29$, $p < 0.001$) and a positive association with education level ($r = 0.14$, $p < 0.05$). Age was positively correlated with the total number of diagnosed health conditions ($r = 0.17$, $p < 0.05$). However, in terms of gender, no significant relationship was detected between gender and SF. Furthermore, there was no significant relationship between the total number of diagnosed health conditions and SF. Similarly, no significant relationship was found between individual chronic diseases (diabetes, hypertension, high cholesterol, hearing problems, vision problems, gout, kidney problems, and others) and SF. The correlation results are presented in Table 3.

**Table 3.** Correlations between the variables ($N = 200$).

| | Age | Education Status | Numbers of Diagnosed Health Conditions | Self-Rated Health Status |
|---|---|---|---|---|
| SF | 0.11 | −0.09 | 0.02 | −0.19 ** |
| Age | - | −0.02 | 0.17 * | 0.09 |
| Education Status | | - | −0.04 | 0.14 * |
| Numbers of Diagnosed Health Conditions | | | - | −0.29 ** |

\* $p < 0.05$ (two-tailed). \*\* $p < 0.01$ (two-tailed).

### 3.3. Frequencies of Chronic Diseases across Social Frailty Levels

Among Hong Kong older adults, the top three chronic diseases were high cholesterol (71, 35.5%), hypertension (61, 30.5%), and visual problems (43, 21.5%). In terms of social frailty levels, SPF had the highest prevalence among individuals with high cholesterol, hypertension, visual problems, and diabetes, followed by SF and SNF. Table 4 presents the detailed frequencies of chronic diseases across the social frailty levels.

**Table 4.** Frequencies of chronic diseases across social frailty levels (*N* = 200).

| Chronic Diseases | SNF | | SPF | | SF | | Total | |
|---|---|---|---|---|---|---|---|---|
| | *N* = 83 | % | *N* = 69 | % | *N* = 48 | % | *N* = 200 | % |
| Diabetes | 6 | 7.2 | 10 | 14.5 | 4 | 8.3 | 20 | 10 |
| Hypertension | 23 | 27.7 | 23 | 33.3 | 15 | 31.3 | 61 | 30.5 |
| High Cholesterol | 28 | 33.7 | 26 | 37.7 | 17 | 35.4 | 71 | 35.5 |
| Hearing Problems | 6 | 7.2 | 5 | 7.3 | 2 | 4.2 | 13 | 6.5 |
| Visual Problems | 16 | 19.3 | 18 | 26.1 | 9 | 18.8 | 43 | 21.5 |
| Gout | 1 | 1.2 | 4 | 5.8 | 0 | 0.0 | 5 | 2.5 |
| Kidney Problems | 4 | 4.8 | 3 | 4.4 | 1 | 2.1 | 8 | 4 |
| Others | 12 | 14.5 | 8 | 11.6 | 8 | 16.7 | 28 | 14 |

Social non-frailty (SNF), social pre-frailty (SPF), social frailty (SF).

*3.4. Frequencies of Social Frailty Questions across Social Frailty Levels*

From the social frailty question results, 81% of older adults have someone to confide in, 77% do not have financial limitations in paying for their required medical services, and 76% do not live alone. The three lowest activities among older adults in the SPF and SF groups were 'sometimes visiting friends' (52.2%, 16.7%), 'turning to family or friends for advice' (56.5%, 20.8%), and 'eat with someone at least once in a day' (59.4%, 25%). Table 5 presents the detailed frequencies of social frailty questions across the social frailty levels.

**Table 5.** Frequencies of social frailty questions across social frailty levels (*N* = 200).

| Social Frailty 8 Questions | SNF | | SPF | | SF | | Total | |
|---|---|---|---|---|---|---|---|---|
| | *N* = 83 | % | *N* = 69 | % | *N* = 48 | % | *N* = 200 | % |
| 1, Do you sometimes visit your friends? (yes) | 74 | 89.2 | 36 | 52.2 | 8 | 16.7 | 118 | 59 |
| 2, Do you turn to family or friends for advice? (yes) | 78 | 94 | 39 | 56.5 | 10 | 20.8 | 127 | 63.5 |
| 3, Do you have someone to confide in? (yes) | 82 | 98.8 | 58 | 84.1 | 22 | 45.8 | 162 | 81 |
| 4, * Do you go out less frequently compared with last year? (no) | 77 | 92.8 | 54 | 78.3 | 17 | 35.4 | 148 | 74 |
| 5, Do you eat with someone at least one time in a day? (yes) | 74 | 89.2 | 41 | 59.4 | 12 | 25.0 | 127 | 63.5 |
| 6, * Are you limited by your financial resources to pay for needed medical service? (no) | 78 | 94 | 52 | 75.4 | 24 | 50.0 | 154 | 77 |
| 7, * Do you live alone? (no) | 76 | 91.6 | 54 | 78.3 | 22 | 45.8 | 152 | 76 |
| 8, Do you talk with someone every day? (yes) | 77 | 92.8 | 52 | 75.4 | 17 | 35.4 | 146 | 73 |

* Reverse scoring questions.

**4. Discussion**

This cross-sectional survey investigated the prevalence of social frailty in Hong Kong. To our knowledge, this is the first study to collect data on social frailty using a tool that has been used in some other international and regional studies [22]. Our study found a higher combined prevalence of socially pre-frail and frail older people of 59%, as compared to a Singapore study, in which the prevalence was only 36% [21]. While both studies had a higher proportion of non-frail older adults, the Hong Kong sample signals a more pessimistic status and increased prevalence of those who are at risk of developing social frailty. The increased prevalence of pre-frail older adults has implications for the next few years of ageing individuals in Hong Kong. The trend of pre-frailty progressing into a frailty state is established [23] and, within the context of social frailty, could lead to permanent physical disabilities within the next 2–3 years [7]. While cross-country comparisons of social frailty research are useful for understanding this social phenomenon, which is common in older people, it is noteworthy to consider the population and sample characteristics that might have a significant influence on the prevalence and social frailty status. These

characteristics encompass both demographic characteristics and socio-economic status, which can also be attributed to the culture and values of older people in Hong Kong.

This cross-sectional survey also examined social frailty status and self-rated health, which suggested that those who belong to the pre-frail and socially frail groups have lower perceptions of their own health status. These findings, although self-reported, have highlighted the strong tendencies of older people with a lower perception of health status to be socially frail, which might have been compounded by the impact of living with chronic conditions. A Japan-based study that examined social frailty in older adults with cardiovascular conditions found how disengagement from meaningful activities due to physical limitations leads to social frailty [24].

In terms of chronic conditions, our study did not show a relationship between social frailty and the total number of chronic diseases or having a specific chronic disease. Several studies have proven the association between chronic diseases and frailty (physical frailty), but there is limited evidence on the relationship between social frailty and chronic conditions. For example, studies have examined the relationship between social frailty and Chronic Obstructive Pulmonary Disease (COPD) and heart failure [25,26]. This study revealed that patients with COPD and HF have low levels of social participation, hence becoming socially frail. However, it was noteworthy that the chronic conditions collected in our study may have a lesser impact on physical activities as compared to those individuals with pulmonary and cardiovascular issues, who normally experience a decline in physical ability that affects social activity participation. From this study of Hong Kong older adults, the top three chronic diseases were high cholesterol, hypertension, and visual problems. In terms of levels, those with SPF status had the highest prevalence among individuals with high cholesterol, hypertension, visual problems, and diabetes, followed by SF and SNF. Although the analysis did not show a relationship between social frailty and chronic disease, the data distribution illustrated that SPF and SF have a higher rate of some chronic diseases than SNF.

In terms of the activities that characterized social frailty in the SF-8 scale, 81% of older adults have someone to confide in, 77% do not have financial limitations in paying for their required medical services, and 76% do not live alone. These findings show that the majority of older adults in Hong Kong in our study cohort may have more social support (friends, family, helpers) and are financially stable enough to afford healthcare. On the contrary, the three lowest activities for those in the SPF and SF groups were 'visiting friends', 'turning to family or friends for advice', and 'eat with someone at least once in a day'. While a majority of older adults in this Hong Kong study appeared to have strong social support and financial stability, those identified as pre-frail and socially frail, particularly those aged 60 and over, showed lower levels of social interaction and support. This highlights the importance of fostering social connections and ensuring financial stability in mitigating social frailty among older people in Hong Kong. Hong Kong is a highly developed region with a robust medical system, a well-educated population, and a strong emphasis on dietary care [27]. However, it also has a high cost of living and a fast-paced lifestyle [28], which could potentially lead to increased stress and reduced time for social interactions, thereby contributing to social frailty. Additionally, the high population density could lead to a lack of personal space, further exacerbating feelings of social isolation due to the lack of opportunities and physical spaces facilitating older people to socialise. Unique to Hong Kong is the presence of a group of caretakers known as 'domestic helpers', whose role in mitigating social frailty could be significant.

## 5. Limitations

The limitation of this study is the self-reported nature of measuring social frailty, which is a common restriction in some cross-sectional surveys of the older population. However, because reference was made to earlier studies on the topic, which also provided a point of comparison, the limitations of the self-reported nature of this study have been addressed, and the results can be considered robust and valid. Moreover, the study sample

was sourced from a database comprising individuals who are members of this group and are accustomed to participating in online surveys, leaving out the cohort who might not have the resources and capacity to participate through this method. It is important to note that these individuals may be more active and more inclined to participate in research compared to the broader older adult population in Hong Kong. Consequently, they may not provide a comprehensive representation of the entire older adult demographic in the region.

## 6. Conclusions

This cross-sectional survey on social frailty among older people in Hong Kong revealed a higher prevalence of pre-frail and socially frail individuals compared to previous studies conducted in other regions. Our findings indicate a higher prevalence of social frailty in Hong Kong; it is essential to delve deeper into these unique factors to fully understand this social phenomenon, which could inform future interventions. Future research should focus on these areas to provide a comprehensive understanding of social frailty in the context of Hong Kong.

**Author Contributions:** Conceptualization, J.M.; methodology, K.K.; software, K.K.; validation, K.K. and K.M.C.L.; formal analysis, K.K. and K.M.C.L.; investigation, K.K.; resources, K.K.; data curation, K.K.; writing—original draft preparation, K.K.; writing—review and editing, K.K. and J.M.; visualization, K.K.; supervision, J.M.; project administration, J.M. All authors have read and agreed to the published version of the manuscript.

**Funding:** This research received no external funding.

**Institutional Review Board Statement:** Ethical approval was obtained from the Institutional Review Board at the Hong Kong Polytechnic University before the study was conducted (Application Number: HSEARS20231011007, Date: 12 October 2023).

**Informed Consent Statement:** Informed consent was obtained from all subjects involved in the study.

**Data Availability Statement:** The raw data supporting the conclusions of this article will be made available by the authors on request.

**Conflicts of Interest:** The authors declare no conflict of interest.

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
