# Peer review of "Social Frailty Prevalence among Older People in Hong Kong"

_2673-9259, doi:10.3390/jal4030016_

Round 1

Reviewer 1 Report

Comments and Suggestions for Authors

 First of all, many thanks to the editor for allowing me to revise the article. It’s always a pleasure to collaborate with this journal. As regards the article, I read the article with interest since the topic is particularly relevant given the gradual increase in the elderly population expected in the coming years. In addition, my interest was also stimulated by the geographical context, the city of Hong Kong, which is characterized by a very large population. Despite my interest, I found the article somewhat naive and superficial in its arguments, and the English should be greatly improved. While reading it, I thought several times about rejecting the article, but in the end I want to give it a chance. In the following paper the authors find some suggestions to improve the article. Apart from the points contained in the paper, I suggest that the whole structure of the article should be better argued, both the methodological part (especially the type of sample) and the conceptual part, by including precise and documented definitions of the main concepts on which the article is based, e.g. social frialty. The authors provide a definition (lines 51-58) but I found it too generic and not adequate for a scientific articles. Finally, I find that all the sections into which the article is divided should be better argued and integrated. Hoping to have given useful advice to the authors to improve the article, I wish them good work.

1.       Lines 22-23, abstract section: I suggest to delete the text “in Hong Kong compared to other regions” and to rephrase the sentence as follow:“This study also indicates a higher incidence of social frailty particularly those in the pre-frail stage, which is  one step closer to the socially frail state.” I suggest changing this sentence as the abstract don’t argument the comparison between Hong Kong and other cities.

2.       Keywords: I suggest to add “Hong Kong”

3.       Lines 33-34: add references to the end of the first sentence

4.       Lines 46-47: I suggest the authors to provide some concrete examples of the social aspects referred to the social frailty

5.       Lines 51-54: add references

6.       Lines 54-56: add references

7.       Lines 57-59: I suggest the author to be more specific in their description of the factors that contribute to a socially frail status. Moreover, what do the authors mean by “limited support”? Is limited support formal or informal, or both? And what do they mean by formal and informal?

8.       Lines 60-61: the sentence is not clear. What do the authors mean for “social frailty has been related to challenges linked to family dynamics”? A more clear description of this concept is expected.

9.       Line 65: what do the authors mean by “participate less in activities”? What kind of activities? Please specify the meaning of “activities”

10.   Lines 70-71: add reference to the end of the sentence

11.   Lines 77-80: I suggest the author to better argument the lack of social frailty studies in Hong Kong. Moreover, I suggest to add the main research questions to the end of the introduction.

12.   Lines 83-86: I suggest improving the research design section with a more detailed description of conceptual framework adopted and/or theories that allowed authors to connect health profile characteristics to social frailty levels. In lines 83-84 an explanation of the cross-sectional quantitative design and of the period of time is expected.

13.   Paragraph “Participants, data collection, and procedures”: a more detailed description of these subjects is needed. In particular, given that this is a cross-sectional study, authors have to add the data collection period (star- end time) and the characteristics of the sample: Is this a convenience sample? Or is it a representative one? Authors must specify the type of the sample, as this determine the generalization of the findings. If the study is based on a convenience sample, the further step is to replicate the research with a representative one. Moreover, since the questionnaire is intended for the elderly, it would be necessary to specify how it was filled out, whether self- administred or or whether the filling out supported by someone. If the latter, who gave support?

14.   Line 96: add a brief presentation of the main sections of the questionnaire, as the reader has to know the questionnaire structure.

15.   Line 97: please rename the paragraph as follow “Demographic Characteristics” and describe in detail the variables related to this topic. I guess you asked for more than 4 structural variables...they are married, widow or other? Do they have children? And what about their income? I think that these variables are important for measuring/analyzing the following Social frailty scale. Furthermore, add a paragraph in which authors describe the questions included in the section focused on the health of the respondents. In particular, in my opinion the 8 items used for assessing health conditions need to be improved, by adding other types of pathologies, such as cognitive impairment, physical impairment, ect.

16.   Lines 107-112: I suggest improving the description of Social Frailty Scale-8 items, by including the subjects investigated in this scale in a more detailed way. At this aim, I suggest to add a table that describes all the items included in the scale; if the items presented in the table 4 are exhaustive, you can add the same table in this section. Moreover, I suggest the author to motivate and argument their choice of this scale. In particular, why is there no consensus on how to accurately measure and assess social frailty? Please, describe the state of the conceptual debate in which the authors' choices matured

17.   Lines 140-143: this is redundant; the text of this question is reported above (see lines 99-102). In the section results, authors have to describe findings, the questions/variables should be described in the section “methods”

18.   Table 4: In my opinion, these results are too generic and need to be described in a more detailed way. At this aim, I suggest the author to argue more thoroughly about the demographic variables of the respondents and the characteristics of the scale (see point 15-16 above) in order to improve the results section.

19.   Line 186: I suggest to tone down this sentence by replacing the verb “examined” with the verb “explored” as this study is not based on a representative sample and the findings need to be deepened with larger studies both in sample size and in number of variables considered. In my opinion, this study is a first glance of this complex phenomenon.

20.   Lines 196-201: these sentences are not so clear, I suggest the authors to better clarify their point of view. Morevoer, what do the authors mean by the sentence “These characteristics en-199 compass both demographic characteristic as well as socio-economic status, which can also 200 be attributed to the culture and values of older people in Hong Kong.”? What would be the values and culture they mention? (see lines 199-200) In a scientific article It is important to be clear and exhaustive in the description of findings and results and you cannot mention a cultural factor without a clear explanation of this.

21.   Line 215: which study are you talking about? Your study? Or the studies reported in ref 17-18? Your study does not include COPD and heart failure among items for assessing chronic disease (see table 3). Please be more detailed in your argumentation, especially in the results section.

22.   Lines 239-249: please add “limitations” in a separate a paragraph (named “limitations”) and add:

-          Name of the dataset used for the sample

-          Type of the sample (convenience or representative) as this characteristics allow to understand the findings generalization

In lines 243-248 the authors mentioned the risk of a bias in the sample strategy, in particular a self-selection bias. The comprehensive representation of the population is mainly due to the type of sample that could be further affected by a self-selection bias. Please be more detailed in reporting the limitations of the study.

23.    Lines 250-266, section Conclusions: I find this section too generic and I read some new concepts, for example fast-paced lifestyle of the Hong Kong population or the presence of 'domestic helpers', that should be descripted in a more detailed way and introducing also in other paragraph, for example “introduction” and “results” in order to better contextualize the study

Comments on the Quality of English Language

I suggest to improve the quality of English by using a proof reading service.

Author Response

Please see the attachment, thanks.

Quality of English Language: We have reviewed the manuscript throughout and consulted a native English speaker for proof reading. Moreover, the first author’s first language is English.

Reviewer 2 Report

Comments and Suggestions for Authors

This article needs a great deal of work. The abstract needs to be rewritten due to lack of clarity. I suggest removing the discussion section from the abstract.

I am unclear on dividing the study into three categories since the focus was mainly on social fragility, and there were no comparisons between the three categories. I suggest including a thorough review of the literature to identify whether social prefrailty and social nonfrailty contribute to the purpose of the study.

Comments on the Quality of English Language

The quality of the English language is appropriate.

Author Response

(The authors gave the same response as above.)

Reviewer 3 Report

Comments and Suggestions for Authors

The study of social frailty among the elderly is becoming a priority task of modern science in connection with the global aging of the population in developed countries.  The authors of this work attempted to establish a connection social frailty with various health characteristics and socio-demogaphic indicators.  It is necessary to note the importance of using health variables in this perspective of the study. However, there are several questions regarding the organization of the article and the evidence contained therein. First, all statistical processing procedures should be presented in the text. How do the indicators on the integral scale of social frailty relate to the scale of normal distribution? It is also necessary to assess the asymmetry and excess of each question, to demonstrate the internal links between the questions (to demonstrate the Cronbach's Alpha indicators). It would also be desirable to demonstrate the results of a confirmatory analysis, according to which all questions are included in one common factor. In addition, the division into 3 levels also requires statistical refinement in accordance with the sample average and deviation from the mean (sigma). The results would be more convincing if the authors conducted a pairwise comparison of averages using the criterion of differences.

I wish success to the authors in publishing the research results!

Author Response

Please see the attachment, thanks.

Quality of English Language: We have reviewed the manuscript throughout and consulted a native English speaker for proofreading. Moreover, the first author’s first language is English.

Round 2

Reviewer 2 Report

Comments and Suggestions for Authors

The manuscript is generally well- written and clear but needs minor revisions. When reviewing the full paper, I have a few comments:

Line 18 -28 - The conclusion in the abstract need to be condense and congruent with the conclusion at the end of the manuscript.  Further, the conclusion of the abstract need to be concise and should not be longer than the results section. In addition, the result section in the abstract need to include more data.

There are a few grammatical errors and the manuscript need to be reviewed for grammatical errors. Consider using a tool to check for grammatical errors.

Comments on the Quality of English Language

Author Response

Please see the attachment, thanks.
